# Exploring the roles of men and masculinities in abortion and emergency contraception pathways, Ghana: a mobile phone-based mixed-methods study protocol

Joe Strong 

Department of Social Policy, London School of Economics and Political Science, London, UK

**Correspondence to**
Joe Strong; j.strong3@lse.ac.uk

## ABSTRACT

**Introduction** Global commitments have established goals of achieving universal sexual and reproductive health and rights (SRHR) access, but critical obstacles remain. Emergency contraception and induced abortion are overlooked in policy and research. Men's roles in the SRHR of others are significant, particularly as obstacles to universal SRHR. Evidence on gender, masculinities and SRHR is essential to understand and reduce the barriers faced by individuals seeking to avoid the conception or continuation of a pregnancy.

**Methods and analysis** This study aims to understand men's masculinities and their relationships with emergency contraception and abortion. The protocol presents a multimethod study of men aged over 18 years in James Town, Accra, Ghana. In response to the COVID-19 pandemic, the research will use two mobile-based methods: a survey and in-depth interviews. Using respondent-driven sampling, an estimated 789 men will be recruited to participate in the survey, asking questions on their knowledge, attitude, behaviours and roles in emergency contraception and abortion. In-depth interviews focused on constructions of masculinity will be conducted with a purposive sample of men who participated in the survey. Data will be analysed concurrently using multiple regression analyses of quantitative data and abductive analysis of qualitative data.

**Ethics and dissemination** Ethical approval has been granted by the London School of Economics and Political Science and the Ghana Health Service. The findings in this study will: engage with emerging research on masculinities and SRHR in Ghana and elsewhere; offer methodological insight for future research; and provide evidence to inform interventions to reduce obstacles for emergency contraception and abortion care seekers. Dissemination will occur at all levels—policy, academic, community—including multiple academic articles, policy briefs, workshops and presentations, conference papers, and theatre/radio-based performances of key messages.

## INTRODUCTION

Postcoital pregnancy avoidance methods, the most effective of which are emergency contraception and induced abortions, are

---

### Strengths and limitations of this study

► Transferring methods to mobile phones allows for research to be conducted in changing environments due to COVID-19, where movement and social contact are problematic.
► A mixed-methods approach strengthens the ability for the study to unpack critical constructions that relate to emergency contraception and induced abortion and increases the richness of the data.
► Mobile-based methods are limiting in terms of only reaching those with access to the technology.
► Mobile phones reduce non-verbal cues and lack of observable confirmation of privacy and consent do not allow for the inclusion of participants who have gatekeepers (eg, young people).

---

under-represented in global and regional policy and research.[1 2] Despite critical international milestones such as the International Conference on Population Development and Sustainable Development Goals 3 and 5[1] and regional commitments (eg, Maputo Protocol[3]) calling for universal access to sexual and reproductive health and rights (SRHR), political, social and economic barriers have restricted progress, particularly in relation to postcoital pregnancy avoidance.[1 4–8]

Emergency contraception is most frequently provided in pill form. Though there is a copper IUD emergency contraceptive, evidence from high-income countries indicates extremely low uptake.[9] There are various types of emergency contraception pills (ECPs) with varying efficacies—the most effective estimated to prevent up to 95% of pregnancies if taken within 5 days of sex that would lead to conception.[10] Abortions are essential healthcare, categorised by the WHO as safe, less safe and least safe.[11] The safety levels depend on the method choice and

the environment where the abortion occurs. However, the advent of medical abortions—misoprostol/mifepristone combinations—has radically reshaped conceptions of safety, with pregnant people able to induce abortions safely outside of facilities, delineating the conflation of safety and overmedicalisation.[11 12]

Abortions and emergency contraception are both legally permissible in Ghana.[13 14] ECPs have been included in health policy since 1996.[15] Knowledge of ECPs is increasing; the 2008 Ghana Demographic Health Survey found 35.4% of women had heard of emergency contraception when asked.[16] More recently, Performance Monitoring for Action (PMA2020) data estimated that 52% of women aged 15–49 had heard of emergency contraception.[17] ECPs are part of the mix of contraceptives that women in Ghana use,[18] with one-third of young women attending pharmacies in Greater Accra reporting ECPs as their main method of contraception.[19]

Abortions have been decriminalised in Ghana since 1985 for three criteria: fetal abnormality; maternal health; and rape/incest.[20] However, knowledge of the law remains low,[21] with only 11% of survey respondents in the 2017 Ghana Maternal Health Survey knowing abortions were legally permitted.[22] Estimates of the abortion rate in Ghana vary from 13.4 per 1000 women aged 15–49 in 2017,[22] to 26.8 per 1000 in the same year,[21] many of which are less and least safe, with significant associated risks to pregnant people's health.[23 24]

Globally, studies since the 1990s have shown that men have direct and indirect roles in emergency contraception[25–28] and abortion-related care.[29–37] Research on emergency contraception and men's roles in Ghana has been limited.[18 19] Abortion-related studies in Ghana indicate that men insert themselves in the autonomous decision-making of others via their role as partner or father,[38 39] including controlling the finances for seeking care,[39] and providing medicines and materials for care (eg, procuring methods).[35 40] In addition, men's responses to (potential and confirmed) pregnancies,[41] including denial of pregnancy,[42] have a significant role in a person's postcoital pregnancy avoidance decision-making. While evidence indicates that men do support abortions, for example, in instances of birth spacing and to finish school, such support continues to decentre pregnant people from their own autonomous reproductive decision-making.[39]

Few studies analyse the mechanisms that drive men's roles in reproduction.[43 44] Critical studies of men and masculinities (CSM) has emerged as a response to this. CSM examines masculinities and femininities in order to understand men as gendered individuals and provide an explanation of the mechanisms that drive social behaviours.[45–47] CSM draws heavily on the notion of 'hegemonic masculinity', which is the contextually idealised form of masculinity.[48 49] Hegemonic masculinities must be understood in the context of intersectional power relations, in which race, nationality, sexuality and ability all impact its construction.[50 51] Complex systems of power mean that masculinities might be hegemonic in

one context but made marginal in another.[51–55] In Ghana, the destructive role of colonialism shaped masculinities by imposing Euro-American normative values, resulting in a complex system of masculinities embedded within a globalised white supremacist power system.[56]

Constructions of masculinities are not static, and new forms emerge and become intertwined in social structures—such as health and legal systems—and used to justify social control and dominance over other forms of masculinities and femininities.[57 58] Understanding masculinities is a mechanism through which to explore power and the potential ways in which these constructions operationalise power over reproductive decision-making. Investigating the characteristics of hegemonic and relational masculinities allows a better understanding of men's roles in postcoital pregnancy avoidance. As a constantly evolving construct, greater evidence on how masculinities operate presents the opportunity to consider how men's masculinities can be engaged with in policy and rights-based activism.

A note on terminology: 'Postcoital pregnancy avoidance' is evolving terminology. It was important to indicate that this research would not be focused on methods of contraception that could be used before or during intercourse (eg, intrauterine devices (IUDs) or condoms). Instead, the focus is methods that occur 'post-coital', particularly after a sexual encounter that risks pregnancy. Emergency contraception and abortion are separate but linked forms of reproductive healthcare. Emergency contraception prevents a pregnancy from occurring, whereas abortions are healthcare for people already pregnant. This distinction is important. 'Avoidance' was used as an abbreviation of these methods that avoid either the conception or the continuation of a pregnancy.

## METHODS/DESIGN
### Aims and objectives
This protocol outlines a funded, multiyear research project, expected to finish in March 2022. The aim of this research is to generate extensive data exploring the relationships between masculinities and postcoital pregnancy avoidance in a study site in Accra, Ghana. It has the following objectives:

1. To identify and explore the factors associated with men's knowledge, attitudes and behaviours towards postcoital pregnancy avoidance.
2. To generate and analyse new knowledge on how men's constructions of masculinities, sex and abortion interact with abortion-related care pathways.
3. To investigate the relationship between men and pathways to obtaining ECPs and to explore the constructions of masculinities within these.

Abortion and ECPs, while linked, are kept distinct in the objectives. This is to reflect the very different purposes each has within a person's reproductive life course, as well as variations in current knowledge and evidence about the two.

Data will be collected through a multimethod, concurrent design. A mixed-methods approach is methodologically most appropriate to achieve these aims and objectives, allowing for both breadth of information through a larger survey and depth of information through qualitative interviews.[59 60] These data will be used to produce substantial outputs, including peer-reviewed papers, blogs, policy briefs and art.

The methods will be triangulated by looking at how masculinities are constructed and relate to abortions within each method and between each method, exposing complexities and narrative devices[61] that will allow for a generative exploration of the objectives.[62]

Objective 1: Quantitative data on sources of knowledge and the association between knowledge, attitudes and behaviours, and sociodemographic factors will be analysed alongside an interrogation of qualitative data on the emergent mechanisms that drive these, with the intention of using uncovering broad trends and exploring the potential mechanisms behind these.

Objectives 2 and 3: Interview data will explore the constructions and operationalisation of 'manhood' or 'masculinities' within discussions of sex, emergency contraception and abortion. Emergent themes will then be used to interrogate the responses men provide when asked about masculinities within the survey and to unpack any broader trends around the construction of masculinities and emergency contraception and abortion pathways, respectively. The data are designed to allow for an exploration of the links in surveys between men's involvement in emergency contraception and abortion pathways, their constructions of their masculinities and the mechanisms emergent in qualitative data that might explain these.

## Formative research and study site selection

This project had two phases of formative research: a scoping trip in January 2019 and a piloting phase in February to March 2020. The formative phase involved conversations with expert professionals working in SRHR, particularly abortion, including activists, advocates, academics and practitioners.[63] Such conversations resulted in the inclusion of ECPs into the research project. An important component of the scoping trip was to identify a relevant area to locate the study.

James Town, a neighbourhood in Accra, was chosen as the study site. Expert conversations indicated that it would be an area where the research topic could be useful for local activists as well as the Ghana Health Service and key stakeholders. Key stakeholders in the community, including the Secretary to the Paramount, a key figurehead in the Ga community in James Town, expressed approval for the research to be conducted.

The majority of residents are ethnically Ga, with largely internal migration meaning that there is an increasing mix of cultural representation from across Ghana.[64] Ga communities have been historically patrilineal, compounded by the imposition of patriarchal British values during colonialism. While relations between genders afford women certain autonomies, pregnancy and childbirth remain important and are linked to men's performance of masculinities.[57] The changing dynamics of the area, for example, from economic internal migration from elsewhere in Ghana,[64] allows for explorations of how constructions of masculinities might vary within and between groups of men.

Access to healthcare remains inequitable, with the Accra Metropolitan Assembly reporting 'inadequate coverage' across of the entire submetro.[65] This includes challenges in both accessing healthcare and in public health, with recent incidences of diarrhoeal disease[66] and cholera[67] outbreaks. Despite a number of maternity-based clinics in James Town,[68] the largest of which is Ussher Polyclinic, it was observable that abortion services are not advertised. Evidence indicates that rates of less and least safe abortions remain high, with pregnant people preferring to self-manage their abortions outside of facilities, often with less safe methods such as herbs, toxins and pharmaceuticals.[38] In addition, the study indicated that partners, predominantly men, created barriers to abortion-related care through withholding moral support or financial resources.[38]

Prior to the second formative research phase, three research assistants (RAs) were hired. The researcher position was advertised on social media and through WhatsApp, with a selection of applicants interviewed. Following a week-long paid training workshop on methods and value clarification on SRHR, the three researchers were asked to join the team. All are from the research context and the research tools were iteratively developed as a team.

During the piloting stage, 39 men aged 18 and above were purposively sampled from Ussher Town, a neighbourhood next to James Town. This aimed to reduce the likelihood of these men being resampled during the data collection phase. The survey instrument was iterated based on cognitive interview feedback from participants to ensure questions were understood consistently. The survey and interview guides are available at the project website: https://www.masculinitiesproject.org/.

## Mobile methods

Data collection will be conducted using mobile phones. The original research plan was to conduct the survey and interviews—alongside focus groups—face to face. However, the COVID-19 pandemic, declared 11 March 2020, necessitated a review of these methods. Mobile phones were the most appropriate option due to the high proportion—94%—of shared and individual mobile phone ownership in urban Ghana.[69] The research team can be distanced from respondents and research can feasibly continue in the event of restricted movements.

Evidence suggests that mobile phone-based data collection can provide rich and quality responses, though can require more reflexivity of how the mode might shape the data.[70 71] Interviews and surveys will be administered by the research team over the call. Self-administered surveys

using smart technologies (eg, mobile apps/web-based apps), and audiovisual platforms, were rejected, as they require a level of literacy and internet/data connectivity inappropriate for the context.[38]

## Quantitative survey

### Respondent-driven sampling

Respondent-driven sampling (RDS) was first conceptualised as an alternative to snowball sampling for populations less likely to participate in research, for whom there was no reliable sampling frame to conduct a simple random sample.[72 73] As there are no appropriate existing mechanisms to identify men through social distanced, mobile phone means, RDS was considered suitable. It relies on a peer-referral system in which pre-existing relationships are used to create chains, with the intention being referrals from peer to peer eventually spread to the point that the final respondents are not known to the initial respondents.

The outcomes of interest are the factors associated with men's knowledge, attitudes and prior behaviour relating to postcoital pregnancy avoidance. Independent variables will include a composite socioeconomic index, which combines questions on employment, residence type and resource availability. In addition, a composite score will be created based on responses to questions on the construction of masculinity. This novel approach will attempt to understand masculinities using quantitative data, which can subsequently be compared with qualitative findings.

### Sample size

The sample of interest in this research are men aged 18 and over, who live in James Town. The survey asks respondents their gender in an open-ended format; this research includes anyone reporting that they are a man.

Estimating a sample size is important to provide a useful target minimum for the data to be statistically useful, although sample size estimation is difficult to do accurately for RDS.[74] The most frequently used method for calculating estimated sample size is to take the simple random sample size and multiple by a design effect (*deff*)[75]:

$$n = deff \cdot \frac{P_A(1-P_A)}{\left(se\left(\hat{P}_A\right)\right)^2}$$

$P_A$ represents the proportion of the population of interest. This was taken as 0.73, which is the proportion of men aged 15–59 who believed that men should be involved in some aspect of SRHR.[76] The assumption is made that this holds true for a population of men aged 18+. The standard error (*se*) is set at 0.05.

Debate continues about an appropriate *deff* value for RDS. The initial use of a *deff* value of 2 has been shown to be too low for most RDS studies, and even a revised value of 4 might be lower than necessary.[77] To minimise the risk of having a *deff* value too low for meaningful analysis, this survey assumes a *deff* of 10, as used in recent RDS studies[74]:

**Table 1** Sampling matrix for RDS seeds and interview respondents

| Age group | James Town North | | James Town South | |
|---|---|---|---|---|
| 18–25 | Ga | Non-Ga | Ga | Non-Ga |
| 26–39 | Ga | Non-Ga | Ga | Non-Ga |
| 40–59 | Ga | Non-Ga | Ga | Non-Ga |
| 60+ | Ga | Non-Ga | Ga | Non-Ga |

RDS, respondent-driven sampling.

$$n = 10 \cdot \frac{0.73(1-0.73)}{(0.05)^2} = 788.4$$

The study should reach an estimated 789 individuals to achieve a minimum estimated sample size.

### Seeds and recruitment

Initially, RDS requires 'seeds' who are ideally individuals with known networks within the sample of interest and become the first in the chain of peer referral. The identification of seeds for this project will use the following matrix (table 1):

The decision was made to stratify the seeds based on age, ethnicity (Ga/non-Ga) and location (James Town North and James Town South) within the community in response to existing evidence,[57] and observations and reflections from the research team of the most common factors affecting relationship networks.

Respondents will be compensated via mobile credit transfer of GHC5 (Ghana cedi, currency in Ghana. GHC5.8:US$1. Exchange rate true on 24 June 2020, per www.exchange-rates.org) for responding and an additional GHC2 per referral. Each respondent will be asked to find a maximum of three referrals. Respondents will be asked to provide potential referrals with the mobile number of the RA or provide the RA with the details to then call and arrange an interview time. Each respondent will be given a short code (said verbally and sent via short message service) to pass onto their referral, in order to track the peer-referral process.

### Remote data checks

Mobile methods and the need for physical distancing mean that survey interviews cannot be checked in person by the principal investigator (PI). A random respondent within each complete set of 10 surveys will be selected per RA to be data checked by a data assessor unaware of the original responses.

A selection of questions will be asked to assess for inconsistencies in questions and to ensure that the respondent had been interviewed. Reinterviewed respondents will receive additional compensation of GHC3 for responding to the data check questions.

### Conceptualising men's masculinities

The survey consists of seven sections: sociodemographics, household and socioeconomics, relationships, emergency

contraception, abortion, masculinity and the abridged Washington Group Questions.[78]

The masculinity section was the most complex to design and benefited from the collaborative approach to this research design.[49 79] Through conversations with men during the preparation phases, piloting the survey with men in similar neighbourhoods around James Town, and workshop discussions with research team, a list of factors that were frequently associated with masculinities was created. Men are asked whether they believe that these factors are important for a man their age, as well as asked open-ended questions of what factors make a 'good' man and a 'good' woman. The language of these questions was tested during the formative stages to ensure consistent comprehension.

### Data processing and analysis

Data will be processed onto RDSAT, a software designed specifically for RDS, to map networks and chains and to process the necessary weights. These weights will be identified by a series of questions in the survey that relate to the personal network size (PNS) of the individual.[74 80] The assumption is that the larger the network size, the greater likelihood of the respondent being selected to participate. Thus, weights are inversely proportional to the PNS.

Data will be cleaned and linear and logistic regressions run to explore these associations between the outcomes of interest and the independent variables.

### Qualitative sampling

In-depth interviews will be conducted from a subsample of the survey. This will be a non-probability sample, with respondents selected to populate the same matrix as that used for the seeds. The aim of this is to maximise obtaining a breadth of different experiences and responses. This will take place concurrently alongside the survey and, if necessary, will continue after the survey has completed, to allow for all respondents to be considered as part of the sample.

Sampling will be assumed to be complete once 'saturation' is reached; when no new data are emerging from the selected sample.[60 81] This project understands saturation as when no new themes or codes emerge.[82] As this research aims to be generative and exploratory, aligning approximately with inductive research, such an approach is recommended.[82]

If reaching saturation requires a significant volume of surveys, there may be financial constraints as well as time constraints. Predictive sample sizes are difficult to estimate, though research using inductive approaches to saturation suggests that much can be established in the earliest interviewers.[83] The budget for interviews is not expected to be spent before saturation is reached.

All survey respondents will be asked if they would be willing to participate in an interview at the end of the survey. The PI will select candidates purposively to populate the sample matrix (table 1). Survey responses will

be reviewed to ensure the diversity of individuals within each metric, until data are saturated. RAs will conduct the interviews and respondents will be compensated an additional GHC10.

### In-depth interview instruments

A semistructured topic guide[84] will be used to explore conceptions of masculinities and elaborate on attitudes and behaviours around postcoital pregnancy avoidance. With consent, interviews will be recorded using devices that can plug into the RA's mobile phone, so that the audio is direct to the recorder and not audible for anyone but the interviewer, who will use headphones.

Interviews will be translated and transcribed. Transcribers with experience and fluency in Twi, Ga and English will be hired through an advertised, paid position in the study site. A random selection of 10% of interviews will be back translated by a separate transcriber, to assess for quality. Significant difference in meaning will result in translations and transcriptions being redone by a third transcriber. All documents will be stored on encrypted software and transferred through a secure platform between the PI and the RAs.

### Qualitative analysis

Interviews are expected to generate a substantial and rich volume of data. As such, abductive analysis is appropriate, in which the analysis focuses on new and innovative findings, in order to test the evidence against existing theories of masculinities.[85] Emerging qualitative data will then be analysed concurrently with the outcomes from the survey. Qualitative data will be analysed using the RQDA platform, allowing integrated qualitative and quantitative data analysis on R.[86]

### Participant and public involvement

Key experts (activists, advocates, practitioners, academics working in the field of SRHR) were involved in the scoping phase of this study to elaborate on critical areas that could then be included in the development of the research question.[63] It was during this phase that emergency contraception was discussed as an important emerging contraceptive, warranting inclusion. Participants were included in the piloting phase of this study to provide feedback during surveys on the relevance of questions and to test the consistency of how questions are understood.

Local organisations provided support in the design phase, such as helping disseminate job advertisements and providing informal conversations about specific contextual factors relating to the research. This study intends to collaborate with local arts-based organisations to produce radio/theatre/other arts-based outputs for dissemination and engagement of the research findings.

### DISCUSSION
### Considerations and limitations

There are methodological and practical considerations and limitations to this research. RDS assumes that

respondents will randomly select referrals from their network, and that subpopulations are not isolated to the point of exclusion.[73] In reality, this is not likely to be the case, and while it is argued this is not necessarily a problem because skewed and non-symmetric responses can be weighted accordingly,[73] it may impact the ability to make generalised population inferences.[74]

Individuals without access to mobile technology are likely to be missed, even if they have access to a shared mobile. Certain populations, such as adolescents and people with disabilities that impact their communication over technology, are excluded. This reflects the inability to use non-verbal cues to minimise the survey impact and to assess the role of gatekeepers during the interview.

Evidence indicates that the role of the interviewer has a direct relationship with the quality of answers relating to abortion. Men in this project might wish to provide what they perceive to be socially desirable answers, which are difficult to mitigate. The pilot survey offered little indication that this would be significant, evidenced by the number of men who offered additional information pertaining to 'sensitive' questions on SRHR. It is important, however, that the role of the interview be central in all the data analysis.

Moreover, one of the benefits of mobile phone use is that there is increased confidentiality, as only the respondent can hear the questions,[87] and men have control over when they would like to call and can choose the location they call from. This provides an interesting opportunity to consider the potential balancing of power that mobile phone methods could provide.

## Ethics

Most important is the safety of the researchers and the respondents during the COVID-19 pandemic. This research will not require any face-to-face contact, nor any movement by RAs that could expose them to unnecessary risk. A guaranteed salary ensures that payment will continue regardless of ability to collect data, until the end of the data collection period (estimated September 2020).

Informed consent will be sought verbally from all respondents and recorded by interviewers onto a document. All data will be stored securely on personal devices and transferred to the PI via secure networks, before being destroyed from any non-PI devices.

The topic could also elicit discomfort in triggering experiences relating to sexual and reproductive health. Information and phone numbers to specialist helplines (Marie Stopes Ghana) will be provided to participants, and all participants will be given the contact of the research team, who will be using specific mobile phones and sim cards bought solely for project use, and the GHS Secretariat on their informed consent sheets for any follow-up necessary, relating to the research.

Ethical approval for this project has been granted by the London School of Economics and Political Science (REC ref 000802c). Ethical approval for the amended research was granted by the GHS Ethics Review Committee (GHS-ERC 008/11/19). Approval was also sought and obtained from the GHS Regional Director for Greater Accra, and community stakeholders and leaders.

## Dissemination

This research expects to produce a significant volume of rich data and diverse outputs. These data will be analysed and published into peer-reviewed papers, expected to be a minimum of three (per research aims and objectives) as well as additional papers reflecting on the methodological innovations.

Policy briefs will be created for dissemination among key stakeholders, particularly the GHS. Community dissemination will include written and non-written mediums, in addition to partnering with local arts organisations to produce radio/ theatre-based dissemination.

The research team already has extensive experience in these areas within James Town. As these outputs can be time consuming, this experience, alongside the direct involvement in data collection, will be beneficial and create a more efficient process conceptualising feasible outputs, reliant on the successful application of funds.

**Acknowledgements** Thanks to Samuel Nii Lante Lamptey, Richard Nii Kwartei Owoo, and Nii Kwartelai Quartey and Act for Change for their support and work within Ghana. With thanks to Professor Ernestina Coast and Dr Tiziana Leone for their support and comments on earlier drafts. Thanks to the Department of Social Policy, London School of Economics and Political Science and the Economic and Social Research Council for their support of this research project and to Katy Footman, Rornald Kananura, Dr Samantha R Lattof, Dr Rishita Nandagiri and Dr Laura Sochas for their practical and methodological advice and support.

**Contributors** This research protocol was written in full by JS.

**Funding** This work was supported by the Economic and Social Research Council (grant number ES/P000622/1).

**Competing interests** None declared.

**Patient and public involvement** Patients and/or the public were involved in the design, or conduct, or reporting, or dissemination plans of this research. Refer to the Methods section for further details.

**Patient consent for publication** Not required.

**Provenance and peer review** Not commissioned; externally peer reviewed.

**ORCID iD**
Joe Strong http://orcid.org/0000-0001-8626-4020

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
