## [Reviewer comments · BMJ Open]

ARTICLE DETAILS

TITLE (PROVISIONAL)	Exploring the roles of men and masculinities in abortion and emergency contraception pathways, Ghana: a mobile phone-based mixed-methods study protocol
AUTHORS	Strong, Joe

VERSION 1 – REVIEW

REVIEWER	Melanie Boeckmann Bielefeld University School of Public Health
REVIEW RETURNED	10-Aug-2020

GENERAL COMMENTS	Thank you for this interesting research proposal tackling a sensitive and important topic. I have a few questions and suggestions: Study site “The established relationship between gendered sexual norms and SRHR access⁵⁴ make it an appropriate case study site.” It is not clear to me why this would be especially relevant to James Town: what is the population make-up? Have previous studies at that site shown links between gendered sexual norms and health access? The article you cite here describes cultural attitudes in Ga communities – but your brief paragraph neither indicates that (only) Ga live in James Town nor that these cultural norms are definitely the reason for lack of access. Please expand your justification of study site selection in this paragraph. Formative research: Took place among expert professionals only or also among couples or men? Piloted among which group? Sex ratio among those piloting? Which method is intended to answer which of the three aims? Could you link back to the objectives in the methods section? Can you also add a figure indicating at which stage which method is triangulated with which? You write concurrent yet the interviews are occurring AFTER the survey, no? Qualitative methods: What type of saturation are you looking for? Will it determine the end of sampling or the end of analysis, or both? (https://www.ncbi.nlm.nih.gov/pmc/articles/PMC5993836/)
---

	While the complete sample size cannot be pre-determined, surely you have a time or financial or other resource-related upper limit for interviews you could feasibly conduct and analyze? How do you want to triangulate quantitative and qualitative findings? Participant involvement: Key experts of which backgrounds? Limitations I am missing a discussion about the sensitivity of the project and the potential for socially desirable results or for lack of participation. Is it likely that the participants have access to a quiet private space when conducting the phone interviews? Is there a mechanism in place in case something gets triggered during the interview, such as referral to a counseling service? Or do you intend to only ask about masculinity without alluding to abortion, for instance? That seems unlikely as it would not answer your research question... please be a bit more specific here about the limitations of conducting a project on gender and pregnancy avoidance. Dissemination Considering the amount of time creating theater or radio pieces AFTER analysis would take, could you add a timeline for the whole study? Is there funding to cover all dissemination activities? Consider adding the survey and interview guide as supplementary materials to allow for replicability and for readers of your final results papers to follow what adaptations had to be made.
--	---

REVIEWER	Rachel Scott London School of Hygiene and Tropical Medicine, UK
REVIEW RETURNED	23-Oct-2020

GENERAL COMMENTS	Thank you for the opportunity to review this protocol. The proposed study, which seeks to better understand the relationships between men's masculinities and emergency contraception and abortion, will make a valuable contribution to the literature in this field. The methods proposed and sampling strategy for participant recruitment are appropriate to the research aims, and (although not without limitations, as described by the author) are a sensible approach in the context of the challenges posed for data collection by COVID-19. The multi-method approach has the benefit of not only enabling a richer set of data to be collected, but also allows for triangulation between findings from the quantitative and qualitative components. A strength of the study is the way it examines post-coital pregnancy avoidance as a whole, considering both abortion and emergency contraception – I think these are often viewed separately and that considering them in parallel will provide a much more comprehensive understanding of strategies to avoid pregnancy, particularly considering the importance of (mifepristone and) misoprostol purchased from pharmacies in many contexts globally. I suppose I do wonder whether post-coital pregnancy avoidance is quite the right phrase, since abortion is usually used to end a confirmed or suspected pregnancy rather than avoid
---

(which I would probably interpret as prevent) a pregnancy, but I also don't have a suggestion for a better alternative!

I have some suggestions for edits and clarifications but I recognise the limited word count constrains what changes can actually be made, although I hope that the authors will find them constructive and useful for future work on the study, even if they cannot be incorporated into this protocol paper.

Note – I have used the page numbers on the full PDF, not the word document (i.e. I have used the ones at the top left of each page).

My comments are set out below:

Introduction:

- The introduction sets out a strong rationale for the study and provides all the necessary background.
- P6 line 26, it may be useful to elaborate a little more on the meaning of safety here and contexts where MA is more safe and less safe
- P6 line 32: Could you clarify what you mean by knowledge or what the question asked? Does this refer to knowledge that EC exists, what it is, how it works?
- P6 line 43: In what year (referring to 11% figure)?
- P6 line 53: Could you elaborate or clarify on what is meant by exerting dominance? I also wondered in this section whether there was any evidence of men's more supportive roles, although I recognise the literature around this may be limited, as that will also likely be important for informing service provision or interventions.
- P6 line 18: this last sentence is slightly alone and I think could be expanded on a bit more to bring the key points made in this paragraph into the context of your study.

Methods/design

- The use of mobile methods is an appropriate adaptation to COVID-19, but also has strengths in and of themselves as a research approach. The sample size is appropriate and the sampling strategy using RDS methodology, and its suitability and limitations for the research, and clearly explained. A strength of the study is its investment in involving key experts and stakeholders, and it is clear that that involvement has been clear in shaping the methods but also, importantly, the research questions. In this section it would be useful to state how the qual and quant components will each address the aims and objectives of the research, and elaborate on how the two components will be brought together (i.e. not just that it will be done concurrently and the software use, but how the findings will be used together to generate a richer understanding of the research question) as I think this is a particular strength of this study.
- In objective 1, 'explore' or 'investigate' may be a better term than 'explain', since explain implied a full understanding which may ultimately be unknowable
- Objective 2 is a little vague, could this be revisited to unpack exactly what you mean by 'analyse the interactions' with respect to abortion pathways?
- Similarly, is it possible to be more specific on what you mean by 'significance' in objective 3?
- Finally, why are the objectives around abortion and objectives around ECP different?

	 • P8 line 9: Could you say a bit more about what you mean by “the established relationship between gendered sexual norms and SRHR access” – do you mean this specifically in relation to this study site? • P8 line 36: Could you say a bit more about what you mean by maintaining quality standards – is this in reference to response rates, representativeness, item non-response, social desirability bias, privacy? • P8 line 44: Could you say briefly here what the purpose of the quantitative survey is, with reference to how it will contribute to addressing your aims and objects (same for the qual component later). I also think that your section on outcomes (p11) would be better placed up here, as it helps to understand the sampling and analysis approach, and what information will be collected in the questionnaire. It would also be useful in line 7 to briefly include some information on how the concept of masculinities will be operationalised and ‘measured’. • P11 line 14: Logistic regression not odds regression, I think? Discussion  • A question on the ethics section – line 15 – is it appropriate for respondents to have the mobile number of the RAs? Presumably the RAs will be supplied with separate ‘work phones’ for this project but it may be a significant burden to the RAs if participants are able to call them, especially if they call with questions related to their SRH care or COVID-19. • Perhaps I am misunderstanding and this just needs to be clarified. The dissemination plan is great, particularly as there are clear plans to disseminate the research through a variety of channels and in a variety of formats, locally and to relevant stakeholders, as well as to other academics. As I said – these are mostly suggestions for clarifications or additional information and I realise that they may not be compatible with a constrained word count. Overall, it sounds like a great study and I looking forward to seeing some of the findings!
--	--

VERSION 1 – AUTHOR RESPONSE

Review comment	Response
Reviewer 1 Study site “The established relationship between gendered sexual norms and SRHR access⁵⁴ make it an appropriate case study site.” It is not clear to me why this would be especially relevant to James Town: what is the population make-up? Have previous studies at that site shown links between gendered sexual norms and health access? The article you cite here describes cultural attitudes in Ga communities – but your brief paragraph neither indicates	Thank you for your comments regarding the study site and also the formative research and piloting. I have noted this and comments from Reviewer 2 and rewritten the section to provide more detail and clarity, acknowledging the comments that you have made here. The new section can be found beginning page 6, line 24, section entitled “Formative research and study site selection”

that (only) Ga live in James Town nor that these cultural norms are definitely the reason for lack of access. Please expand your justification of study site selection in this paragraph.	
Formative research: Took place among expert professionals only or also among couples or men? Piloted among which group? Sex ratio among those piloting?	Thank you for this comment on the need for better clarity. I have added in additional details about the demographics of the men in the pilot survey. This can be found in the section “Formative research and study site selection”, paragraph beginning page 7, line 21. Quoted text below: During the piloting stage, 39 men aged 18 and above were purposively sampled from Ussher Town, a neighbourhood next to James Town. This aimed to reduce the likelihood of these men being resampled during the data collection phase. The survey instrument was iterated based on cognitive interview feedback from participants to ensure questions were understood consistently. The survey and interview guides are available at the project website: https://www.masculinitiesproject.org/
Which method is intended to answer which of the three aims? Could you link back to the objectives in the methods section? Can you also add a figure indicating at which stage which method is triangulated with which? You write concurrent yet the interviews are occurring AFTER the survey, no?	Thank you, I have written a short paragraph to explain that the methods will be used across all three objectives [page 6, line 18]: The qualitative and quantitative methods will be used in combination to explore all three objectives. Methods will be triangulated by looking at how masculinities are constructed and relate to abortions within each method and between each method, exposing complexities and narrative devices that will allow for a generative exploration of the objectives. I have also clarified on page 10, line 44, that the qualitative interviews are conducted concurrently at the same time as the survey is ongoing:

	This will take place concurrently alongside the survey and, if necessary, will continue after the survey has completed, to allow for all respondents to be considered as part of the sample.
Qualitative methods: What type of saturation are you looking for? Will it determine the end of sampling or the end of analysis, or both? (https://www.ncbi.nlm.nih.gov/pmc/articles/PMC5993836/) While the complete sample size cannot be pre-determined, surely you have a time or financial or other resource-related upper limit for interviews you could feasibly conduct and analyze?	Thank you, this is a really helpful and relevant resource. I have added more description of how saturation will be conceptualised for this project, page 11, line 5: This project understands saturation as when no new themes or codes emerge. As this research aims to be generative and exploratory, aligning approximately with inductive research, such an approach is recommended. If reaching saturation requires a significant volume of surveys, there may be financial constraints, as well as time constraints. Predictive sample sizes are difficult to estimate, though research using inductive approaches to saturation suggest that much can be established in the earliest interviewees. The budget for interviews is not expected to be spent before saturation is reached.
How do you want to triangulate quantitative and qualitative findings?	Thank you for this comment. I have combined this with the point above regarding the intended 'within' and 'between' analysis of the data from the two methods [page 6, line 18]: The qualitative and quantitative methods will be used in combination to explore all three objectives. Methods will be triangulated by looking at how masculinities are constructed and relate to abortions within each method and between each method, exposing complexities and narrative devices that will allow for a generative exploration of the objectives.
Participant involvement: Key experts of which backgrounds?	Thank you, I have added more detail on page 6, line 27 about the backgrounds of key experts and

	provided a reference to an earlier publication with more information: The formative phase involved conversations with expert professionals working in SRHR, particularly abortion, including activists, advocates, academics and practitioners
Limitations I am missing a discussion about the sensitivity of the project and the potential for socially desirable results or for lack of participation. Is it likely that the participants have access to a quiet private space when conducting the phone interviews?	Thank you, this is important, and I am grateful for this reminder. I have added more detail on page 12, from line 34: Evidence indicates that the role of the interviewer has a direct relationship with the quality of answers relating to abortion. Men in this project might wish to provide what they perceive to be socially desirable answers, which are difficult to mitigate. The pilot survey offered no indication that this would be significant, evidenced by the number of men who offered additional information pertaining to 'sensitive' questions on SRHR. It is important, however, that the role of the interview be central in all the data analysis. Moreover, one of the benefits of mobile phone use is that there is increased confidentiality, as only the respondent can hear the questions, and men have control over when they would like to call and can choose the location they call from. This provides an interesting opportunity to consider the potential balancing of power that mobile phone methods could provide.
Is there a mechanism in place in case something gets triggered during the interview, such as referral to a counseling service? Or do you intend to only ask about masculinity without alluding to abortion, for instance? That seems unlikely as it would not answer your research question... please be a bit more specific here about the limitations of conducting a project on gender and pregnancy avoidance.	Thank you, I have added the plan for responding to any research related questions and impacts on page 13, line 17: The topic could also elicit discomfort in triggering experiences relating to sexual and reproductive health. Information and phone numbers to specialist helplines (Marie Stopes Ghana) will be provided to participants, and all participants will be given the contact of the research team, who will

	be using specific mobile phones and sim cards bought solely for project use, and the GHS Secretariat on their informed consent sheets for any follow up necessary, relating to the research.
Dissemination Considering the amount of time creating theater or radio pieces AFTER analysis would take, could you add a timeline for the whole study? Is there funding to cover all dissemination activities?	Have added information reflecting on the possibilities of dissemination on page 13, line 42: The research team already has extensive experience in these areas within James Town. As these outputs can be time consuming, this experience, alongside the direct involvement in data collection, will be beneficial and create a more efficient process conceptualising feasible outputs, reliant on the successful application of funds. Due to the shifting timelines in the wake of COVID and the inability to accurately create a timeline, as well as limits on wordcount, a final timeline has not been created.
Consider adding the survey and interview guide as supplementary materials to allow for replicability and for readers of your final results papers to follow what adaptations had to be made.	Supplementary materials will be made available when appropriate on the project website - https://www.masculinitiesproject.org/ - which has been added into the text on page 7, line 26.
Reviewer 2	
I suppose I do wonder whether post-coital pregnancy avoidance is quite the right phrase, since abortion is usually used to end a confirmed or suspected pregnancy rather than avoid (which I would probably interpret as prevent) a pregnancy, but I also don't have a suggestion for a better alternative!	Yes, thank you! I completely acknowledge this point – it is a complex term. I have added a footnote onto page 4 of the manuscript that outlines the evolving terminology used, and I have changed the title to specify EC and abortion. The footnote reads: This is evolving terminology. It was important to indicate that this research would not be focused on methods of contraception that could be use pre or during intercourse (e.g. IUDs or condoms). Hence, these are methods

	that occur “post-coital”, particularly post a sexual encounter that risks pregnancy. Emergency contraception and abortion are, however, separate but linked forms of reproductive healthcare. Emergency contraception prevents a pregnancy from occurring, whereas abortions are healthcare for people already pregnant. This distinction is important. “Avoidance” was used as an abbreviation of these methods that avoid either the conception or the continuation of a pregnancy.
Introduction:  The introduction sets out a strong rationale for the study and provides all the necessary background. 	Thank you
P6 line 26, it may be useful to elaborate a little more on the meaning of safety here and contexts where MA is more safe and less safe	Thank you. I have added in more detail relating specifically to medical abortion and safety on page 4, line 17-20: However, the advent of medical abortions – misoprostol / mifepristone combinations – has radically reshaped conceptions of safety, with pregnant people able to induce abortions safely outside of facilities, delineating the conflation of safety and over-medicalisation.
P6 line 32: Could you clarify what you mean by knowledge or what the question asked? Does this refer to knowledge that EC exists, what it is, how it works?	I have changed this from: Knowledge of ECPs is increasing; the 2008 Ghana Demographic Health Survey found knowledge among women at 35.4%. To Knowledge of ECPs is increasing; the 2008 Ghana Demographic Health Survey, found 35.4% of women had heard of emergency contraception when asked. [page 4 line 24]
P6 line 43: In what year (referring to 11% figure)?	I have changed this from: However, knowledge of the law remains low, with only 11% of survey respondents knowing abortions were legally permitted. To:

	However, knowledge of the law remains low, with only 11% of survey respondents in the 2017 Ghana Maternal Health Survey knowing abortions were legally permitted. [page 4, line 32]
P6 line 53: Could you elaborate or clarify on what is meant by exerting dominance? I also wondered in this section whether there was any evidence of men's more supportive roles, although I recognise the literature around this may be limited, as that will also likely be important for informing service provision or interventions.	Thank you for the comment, I have changed the text to be more explicit about how dominance is exerted – see page 5, line 2.
P6 line 18: this last sentence is slightly alone and I think could be expanded on a bit more to bring the key points made in this paragraph into the context of your study.	I have combined with the final paragraph and expanded on it [page 5, line 25 onwards]: Constructions of masculinities are not static, and new forms emerge and become intertwined in social structures – such as health and legal systems – and used to justify social control and dominance over other forms of masculinities and femininities. Understanding masculinities is a mechanism through which to explore power and the potential ways in which these constructions operationalise power over reproductive decision-making. Investigating the characteristics of hegemonic and relational masculinities allows a better understanding of men's roles in post-coital pregnancy avoidance. As a constantly evolving construct, greater evidence on how masculinities operate presents the opportunity to consider how men's masculinities can be engaged with in policy and rights-based activism.
Methods/design In this section it would be useful to state how the qual and quant components will each address the aims and objectives of the research, and elaborate on how the two components will be brought together (i.e. not just that it will be done concurrently and the software use, but how the findings will be used together to generate a richer understanding of the research question) as I think this is a particular strength of this study.	I have written a paragraph linking the methods across objectives and how they will strengthen the study when used together [page 6, line 18]: The qualitative and quantitative methods will be used in combination to explore all three objectives. Methods will be triangulated by looking at how masculinities are constructed and relate to abortions within each method and between each method, exposing complexities and narrative devices that

	will allow for a generative exploration of the objectives.
In objective 1, 'explore' or 'investigate' may be a better term than 'explain', since explain implied a full understanding which may ultimately be unknowable	I have changed this to 'explore' – page 6, line 1
Objective 2 is a little vague, could this be revisited to unpack exactly what you mean by 'analyse the interactions' with respect to abortion pathways?	I have changed this from: To generate new knowledge and analyse the interactions between men, masculinities and abortion pathways. To To generate and analyse new knowledge on how men's constructions of masculinities, sex and abortion interact with abortion-related care pathways [page 6, line 3]
Similarly, is it possible to be more specific on what you mean by 'significance' in objective 3?	I have changed this from: To investigate the significance of men and masculinities in the procurement and use of ECPs. To To investigate the relationship between men and pathways to obtaining emergency contraception pills and to explore the constructions of masculinities within these [page 6, line 5]
Finally, why are the objectives around abortion and objectives around ECP different?	Whilst I agree that the two are linked, the decision was made to keep them distinct in the objectives. This was to acknowledge that they are distinct, and to be careful to avoid confusion of emergency contraception and abortion methods. I have put this into the text [page 6, line 8]: Abortion and ECPs, whilst linked, are kept distinct in the objectives. This is to reflect the very different purposes each has within a person's reproductive life course, as well as variations in current knowledge and evidence about the two.

P8 line 9: Could you say a bit more about what you mean by “the established relationship between gendered sexual norms and SRHR access” – do you mean this specifically in relation to this study site?	Thank you for this comment. I have also received comments relating to the study site from Reviewer 1 and have completely rewritten this section to reflect all of these. Starting from page 6, line 26, the new section is expanded to give more detail in response to your combined comments.
P8 line 36: Could you say a bit more about what you mean by maintaining quality standards – is this in reference to response rates, representativeness, item non-response, social desirability bias, privacy?	I have amended this to: Evidence suggests that mobile phone-based data collection can provide rich and quality responses, though can require more reflexivity of how the mode might shape the data. [page 7, line 39]
P8 line 44: Could you say briefly here what the purpose of the quantitative survey is, with reference to how it will contribute to addressing your aims and objects (same for the qual component later). I also think that your section on outcomes (p11) would be better placed up here, as it helps to understand the sampling and analysis approach, and what information will be collected in the questionnaire. It would also be useful in line 7 to briefly include some information on how the concept of masculinities will be operationalised and ‘measured’.	Thank you, this corresponds to feedback from Reviewer 1. I have created a new paragraph outlining how these methods will relate to the objectives [Page 6, line 18]: The qualitative and quantitative methods will be used in combination to explore all three objectives. Methods will be triangulated by looking at how masculinities are constructed and relate to abortions within each method and between each method, exposing complexities and narrative devices that will allow for a generative exploration of the objectives. I have also added in a section to elaborate more on the section of the survey that covers the survey sections and the conceptualisation of men’s masculinities within the survey. This begins page 10, line 15, with the new text quoted below: The survey consists of seven sections: socio-demographics, household and socioeconomic, relationships, emergency contraception, abortion, masculinity and abridged Washington Group Questions. The masculinities section was the most complex to design and benefited from

	the collaborative approach to this research design. Through conversations with men during the preparation phases, piloting the survey with men in similar neighbourhoods around James Town, and workshop discussions with research team, a list of factors that were frequently associated with masculinities were created. Men are asked whether they believe that these factors are important for a man their age, as well as asked open ended questions of what factors make a 'good' man and a 'good' woman. The language of these questions was tested during the formative stages to ensure consistent comprehension. Finally, I have moved the outcomes of interest to page 8, line 14, in response to your helpful comment about it framing the quantitative survey section.
P11 line 14: Logistic regression not odds regression, I think?	Yes – thank you. Text amended to specify logistic regression [page 10, line 36]
A question on the ethics section – line 15 – is it appropriate for respondents to have the mobile number of the RAs? Presumably the RAs will be supplied with separate 'work phones' for this project but it may be a significant burden to the RAs if participants are able to call them, especially if they call with questions related to their SRH care or COVID-19.	Thank you, I have clarified that all the research team are given mobile phones for specific project use.
The dissemination plan is great, particularly as there are clear plans to disseminate the research through a variety of channels and in a variety of formats, locally and to relevant stakeholders, as well as to other academics.	Thank you

VERSION 2 – REVIEW

REVIEWER	Rachel Scott London School of Hygiene and Tropical Medicine; UK
REVIEW RETURNED	06-Jan-2021

VERSION 2 – AUTHOR RESPONSE

Reviewer comment	Original paragraph	New paragraph
The only point that I feel has not yet been fully addressed is the comment about how the qual and quant components will each address the aims and objectives of the research – you have added the section saying that both components will address all three objectives, but it would be helpful to cross reference with the research objectives to make clear how each component will contribute to addressing each objective. However, I will leave the editor to assess whether this is a necessary further revision.	The qualitative and quantitative methods will be used in combination to explore all three objectives. Methods will be triangulated by looking at how masculinities are constructed and relate to abortions within each method and between each method, exposing complexities and narrative devices⁶¹ that will allow for a generative exploration of the objectives.⁶²	The methods will be triangulated by looking at how masculinities are constructed and relate to abortions within each method and between each method, exposing complexities and narrative devices⁶¹ that will allow for a generative exploration of the objectives.⁶² Objective 1: Quantitative data on sources of knowledge and the association between knowledge, attitudes and behaviours and socio-demographic factors will be analysed alongside an interrogation of qualitative data on the emergent mechanisms that drive these, with the intention of using uncovering broad trends and exploring the potential mechanisms behind these. Objective 2 & 3: Interview data will explore the constructions and operationalisation of ‘manhood’ or ‘masculinities’ within discussions of sex, emergency contraception and abortion. Emergent themes will then be used to interrogate the responses men provide when asked about masculinities within the survey and to unpack any broader trends around the construction of masculinities and emergency contraception and abortion pathways, respectively. The data are designed to allow for an exploration of the links in surveys between men’s involvement in emergency contraception and abortion pathways, their constructions of their masculinities, and the mechanisms emergent in

		qualitative data that might explain these.
--	--	--

VERSION 3 – REVIEW

REVIEWER	Rachel Scott London School of Hygiene and Tropical Medicine; UK
REVIEW RETURNED	19-Jan-2021

GENERAL COMMENTS	Thank you for the additional paragraph, which has really helped to lay out how each objective will be addressed, and how the qualitative and quantitative components will be used together. Looking forward to reading some the outputs of the research!
--